# KURISU-G²: A KNOWLEDGE RETRIEVAL AND GENERATION ALGORITHM BASED ON DOCUMENT STRUCTURE

## ABSTRACT

Retrieval-Augmented Generation (RAG) has become a standard paradigm for enriching large language models with external knowledge, yet it often treats retrieved chunks independently and overlooks their semantic and logical dependencies, leading to incoherent or incomplete answers. GraphRAG addresses this by introducing graph-based context representations, but it remains limited by the quality of the constructed graph, the heavy reliance on LLMs for graph generation, and the lack of global logical consistency. In this work, we propose an alternative perspective: leveraging principled graph-based similarity measures, such as the Gromov–Wasserstein distance, to guide the retrieval, selection, and unification of knowledge units. This approach preserves both the structural and relational properties of the knowledge base, while enabling the enrichment of missing links that are crucial for semantic integrity. We show that this perspective yields more coherent and interpretable retrieval contexts compared to LLM-driven graph construction. Our results highlight a promising path toward robust and logically consistent retrieval mechanisms in RAG-based systems, with strong implications for high-stakes domains such as medicine and law.

## 1 INTRODUCTION

While LLMs are highly effective to answer general queries, their costly training makes it impractical to incorporate domain-specific or up-to-date information. To address this, Retrieval-Augmented Generation (RAG) (Lewis et al., 2020) has emerged as a popular framework that enriches each query with a context based on relevant knowledge units retrieved from a document. However, its retrieval mechanism does not take into account the document structure thus resulting in an often redundant, conflicting, or jumbled context, hence lowering the relevancy of the generated responses. In this paper, we describe a novel mechanism that constructs a more reliable context by taking into account the structural and semantic relationships among knowledge units.

Indeed, RAG typically ranks and selects context chunks independently, based solely on similarity to the query, without considering inter-chunk relationships which may fail to capture deeper semantic or structural connections in the data. The phenomenon of the Lost In Middle (Liu et al., 2023) also reveals that LLMs can struggle finding the key information in a long and redundant context, which deteriorates the quality of the generated answer. These limitations motivate the exploration of more structured retrieval mechanisms that can account for the relationships between retrieved units of knowledge and improve the logical consistency of the response. In particular, graph-based approaches have emerged as a promising direction, aiming to model dependencies and interactions among knowledge units. Indeed, graphs as a data structure are particularly well-suited for representing relationships and dependencies among knowledge units and come with an already established environment and framework. Recent approaches try to formalize the integration of text into a graph structure, leading to Text-Attributed Graph (Zhao et al., 2023; Yang et al., 2021). However,

existing methods such as GraphRAG (Edge et al., 2024) still face challenges in constructing reliable graphs and leveraging them effectively during retrieval. These approaches tackle the lack of coherence and links among the pieces of contexts retrieved by RAG by explicitly modeling the relationships between knowledge units as a graph. They can thus retrieve not only relevant chunks, but also a coherent and connected subgraph of information. However, GraphRAG and similar methods such as ToG-2 (Ma et al., 2024) still suffer from several limitations. Among those limitations lies the construction of the graph itself, which is created with the usage of Large Language Models (LLMs) and is often heuristic as well as computationally expensive. Moreover the choices in the graph traversal also rely on LLMs calls coupled with other pruning heuristics, which can lead to biases in the representation of knowledge. These kind of method are not scalable as they require a massive usage of LLMs to create the graph and to traverse it. Finally, existing graph-based methods often treat the graph as a static structure and do not adapt it dynamically to the specific query or evolving context. These observations motivate the development of more principled approaches to knowledge retrieval that explicitly account for the structure, coherence, and relevance of the retrieved information. Kurisu-G² is an alternative that achieves a trade-off between complexity and the richness of semantic links. It will use a graph structure that will fetch the already existing links in the corpus as a basis to build a more coherent and contextually relevant subgraph using a merging process to combine the relevant pieces of informations as well as a grafting process to add new links between the relevant pieces of information.

## 2 METHODS

The algorithm we propose is based on the idea of exploring a Graph that represents the document containing the content we want to use. Its aim is to identify the most relevant sub-nodes with respect to a given question, using semantic similarities and a structural preservation constraint. In order to do so, it will traverse the graph recursively, and at each depth, it will attempt to merge the relevant sub-nodes while preserving the overall structure. If some attempted fusions do not respect the structural preservation constraint, some smaller deformation of the graph will be allowed, and notably the addition of new edges between the nodes that are not fused. At the end, it will return a path that contains the nodes (or the fused nodes). Depending on the precision regarding the context we need, we can choose the context contained in the node at the end of the traversal which contains less but precise information while the nodes at the beginning of the traversal will contain more information that may not be needed to answer the question.

### 2.1 DOCUMENT GRAPH CONSTRUCTION

The first step in our approach is to construct a document graph that captures the relationships between knowledge units (e.g., sentences, paragraphs, or documents) in the corpus. The aim of this graph is to be a starting point for the retrieval process. Indeed, the core idea of this step is to use the structural links that already exist in the corpus to create a graph whose edges already represent some logical or semantic relationships between the knowledge units. In order to achieve this, we will not use Knowledge Graphs as they are commonly described as a collection of entities and their relationships, but rather we will use a graph where each node will contain a fragment of the text (e.g., a sentence or a paragraph) and the edges will represent the relationships between these fragments. Typically, a directed edge from node A to node B indicates that the fragment in node A is related to the fragment in node B, for example, at the first step of the process, a link between two nodes can be created if the fragment in node B (sentence) is contained in the fragment in node A (paragraph). This first step is thus a parsing step that will use models such as Spacy(Honnibal et al., 2020) to parse the text and extract the sentences, paragraphs, and other relevant information.

## 2.2 FORMALIZATION OF THE STRUCTURE

### 2.2.1 PROBLEMATIC

The problem we are trying to solve concerns the retrieval of knowledge units from a corpus. In order to do so, we need to fuse different knowledge units in our graph so that we catch the most relevant information related to the query. However, the retrieval of knowledge units is not a trivial task. We must avoid retrieving excessive irrelevant information, while ensuring we do not miss key information needed to answer the query. This is where the fused Gromov-Wasserstein distance comes into play, as it will allow us to quantify how the graph is modified in respect to the original graph, and thus to quantify how much information is lost or gained in the process of fusion. The fused Gromov-Wasserstein distance is a distance that can be defined in the space of graphs, and it is based on the idea of comparing the structure of two graphs while taking into account the textual content of the nodes.

### 2.2.2 EVOLVING HIERARCHICAL DOCUMENT GRAPH

We define a **Hierarchical Document Graph (HDG)** as a labeled, directed graph:

$$G = (V, E, \ell_V, W)$$

where:

- $V$ is a finite set of nodes;
- $E \subseteq V \times V$ is a set of directed edges.
- $\mathcal{T}$ is the set of natural language texts.
- $\ell_V : V \to \mathcal{T}$ is a node labeling function assigning to each node $v \in V$ its raw textual content:

$$\ell_V(v) = \tau_v.$$

- $W : V \times V \to [0, 1]$ is the edge existence matrix, where:

$$W(u, v) = p_{uv}$$

represents the weight of existence of an edge from $u$ to $v$, considered during traversal or fusion.

**Additional functions.** We define the following auxiliary functions:

- $\kappa : \mathcal{T} \to \mathcal{T}$, a function that maps raw text to its canonical or filtered version:

$$\overline{\tau} = \kappa(\tau).$$

- $\phi : \mathcal{T} \to \mathbb{R}^d$, an embedding function applied to text:

$$\chi = \phi(\tau), \quad \overline{\chi} = \phi(\overline{\tau}).$$

- $\delta : \mathbb{R}^d \times \mathbb{R}^d \to \mathbb{R}_+$, a semantic or structural dissimilarity between node embeddings:

$$d_{uv} = \delta(\chi_u, \chi_v).$$

**Neighborhood.** The (directed) neighborhood of a node $v \in V$ is defined as:

$$\mathcal{N}(v) = \{u \in V \mid (v, u) \in E\}$$

**Hierarchical content composition.** Initially, the rule we use to create our base Hierarchical Document Graph is the following recursive content aggregation function: $\mathcal{C} : V \to \mathcal{T}$ such that:

$$\mathcal{C}(v) = \kappa(\tau_v) \cup \bigcup_{u \in \mathcal{N}(v)} \mathcal{C}(u)$$

where $\cup$ denotes string concatenation.

**Fusion operator.** Given a set of nodes $\mathbb{F} \subseteq V$, the fusion operator $\text{FUSE}(\mathbb{F}) \to v' \in V'$ constructs a new node $v'$ with:

$$\ell_V(v') = \sum_{u \in \mathbb{F}} \tau_u, \quad \mathcal{N}(v') = \bigcup_{u \in \mathbb{F}} \mathcal{N}(u)$$

The embedding and cleaned content of $v'$ are computed externally:

$$\overline{\tau_{v'}} = \kappa(\ell_V(v')), \quad \chi_{v'} = \phi(\ell_V(v')), \quad \overline{\chi_{v'}} = \phi(\overline{\tau_{v'}})$$

### 2.2.3 Structural dissimilarity definition

In our case, we wanted to weight the edges with the cosine similarity between the embeddings of the content of the nodes, which can be computed as:

$$S_{uv} = \frac{\langle \chi_u, \chi_v \rangle}{\|\chi_u\| \|\chi_v\|}.$$

However, this similarity lies in the interval $[-1, 1]$, and we needed a dissimilarity that lies in the interval $[0, +\infty[$. In order to do so we used the transformation:

$$d_{uv}^p = \left( \frac{1}{\frac{\exp(S_{uv})}{\exp(1)}} \right)^p \quad = \quad \left( \frac{\exp(p)}{\exp(S_{uv} \cdot p)} \right).$$

The interest of such a transformation is to bound the costs of the edges, preventing them from becoming too large if the initial similarity was too close to $-1$

### 2.2.4 Fused Gromov-Wasserstein Distance Between HierarchicalDocument Graphs

Let $G_1 = (V_1, E_1, \ell_V^{(1)}, W_1)$ and $G_2 = (V_2, E_2, \ell_V^{(2)}, W_2)$ be two labeled directed graphs representing two distinct knowledge subgraphs, each following the Evolving Hierarchical Document Graph formalism.

For every node $v_i^{(1)} \in V_1$, $v_j^{(2)} \in V_2$, we compute their embeddings via the external embedding function:

$$\chi_i^{(1)} = \phi(\ell_V^{(1)}(v_i^{(1)})), \quad \chi_j^{(2)} = \phi(\ell_V^{(2)}(v_j^{(2)}))$$

We define the structural dissimilarity matrices $D_1 \in \mathbb{R}^{n_1 \times n_1}$ and $D_2 \in \mathbb{R}^{n_2 \times n_2}$ by computing pairwise dissimilarity between embeddings of nodes within each graph using the external dissimilarity function $\delta$:

$$(D_1)_{ik} = \delta(\chi_i^{(1)}, \chi_k^{(1)}), \quad (D_2)_{jl} = \delta(\chi_j^{(2)}, \chi_l^{(2)})$$

We also define:

- Two discrete probability distributions $\mu_1 \in \Delta^{n_1}$, $\mu_2 \in \Delta^{n_2}$ over the nodes of $G_1$ and $G_2$, typically uniform.

- A content cost matrix $C \in \mathbb{R}^{n_1 \times n_2}$ defined as:

$$C_{ij} = \|\chi_i^{(1)} - \chi_j^{(2)}\|^2.$$

Then the **Fused Gromov-Wasserstein distance** between the two graphs is given by:

$$\mathcal{D}^2_{\text{FGW}}(G_1, G_2) = \min_{T \in \Pi(\mu_1, \mu_2)} \left( \alpha \sum_{i,j} C_{ij} T_{ij} + (1 - \alpha) \sum_{\substack{i,k \\ j,l}} |D_1(i,k) - D_2(j,l)|^2 \cdot T_{ij} T_{kl} \right)$$

where:

- $T \in \mathbb{R}^{n_1 \times n_2}$ is a transport plan with marginal constraints $T \cdot \mathbf{1}_{n_2} = \mu_1, \quad T^\top \cdot \mathbf{1}_{n_1} = \mu_2$.

- $\alpha \in [0, 1]$ balances the influence of node content alignment versus structural consistency.

- The first term accounts for semantic distance between nodes.

- The second term penalizes discrepancies between intra-graph pairwise structural dissimilarities.

This distance provides a principled way to jointly compare the semantic content and structural layout of two knowledge subgraphs, enabling graph alignment, clustering, or fusion tasks under structural constraints.

### 2.3 Recursive Traversal with FGW-Constrained Fusion and Edge Grafting

#### 2.3.1 Algorithm Overview

The algorithm begins by encoding the query $q$ into a vector $\chi_q$ and computing cosine similarities between $\chi_q$ and the children of the current node, which are then ranked in descending order of relevance. The most similar children above a predefined threshold are selected for greedy fusion: if the FGW distance of the fused graph satisfies the constraint, a new node $f$ is created and connected to the explored node $c$ with an edge whose existence weight $p_{cf}$ equals the average weight of the edges linking $c$ to the fused children. Finally, additional edge grafting is performed by evaluating relevant grandchildren not included in the fusion; for each candidate $g$, the similarity $\text{sim}(q, g)$ is computed, and the connection to $f$ is only preserved if it also meets the FGW distance threshold, thus preventing disruptive insertions.

The whole pseudocode of the algorithm is given in the Appendix B.

#### 2.3.2 Edge Evolution

The existence weights of candidate edges evolve through a reward distribution mechanism. At each step, a global reward budget $R$ is distributed among edges according to their similarity relative to the mean of the current neighborhood. Edges with above-average similarity receive proportionally larger increments through a softmax allocation, while edges significantly below the mean are slightly penalized. This evolution rule reinforces semantically coherent links while gradually suppressing irrelevant or noisy ones. A discussion regarding the choice of some concepts of the algorithm and what are the interests of such choices is given in the Appendix B.2.

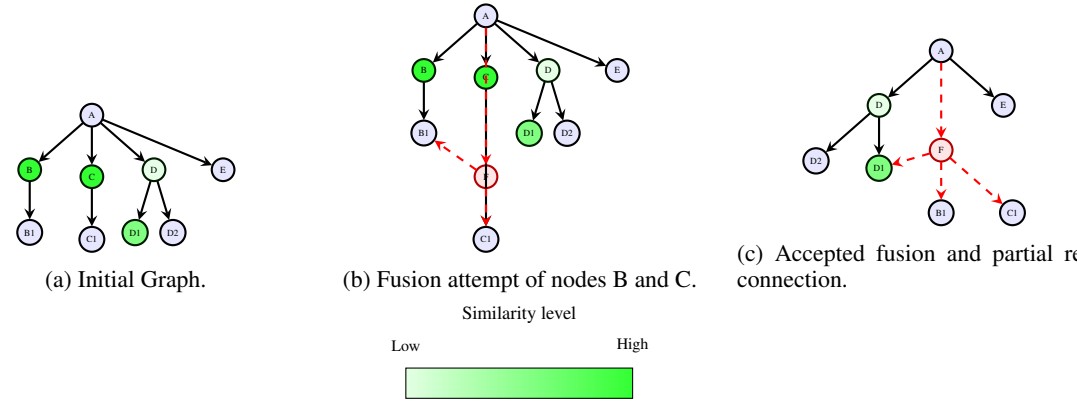

(a) Initial Graph.     (b) Fusion attempt of nodes B and C.     (c) Accepted fusion and partial re-connection.

Figure 1: Steps of the traversal and greedy fusion in the algorithm.

### 2.3.3 THEORETICAL COMPLEXITY ANALYSIS

We analyze the time complexity of the recursive traversal algorithm (v3), which includes greedy fusion and edge grafting, both constrained by a Fused Gromov-Wasserstein (FGW) distance threshold. Let $n$ the number of nodes in the graph, $k$ the average number of successors per node, and $D$ the maximum recursion depth (max_depth). The complexity of computing the FGW distance between two graphs of size $n$ is denoted as $\texttt{FGW}(G_1, G_2)$, which is assumed to be $\mathcal{O}(n^3)$(Titouan et al., 2019).

**Greedy Fusion with Dichotomic Search.** At each depth level, the algorithm selects the relevant children (typically up to $k$) and attempts to find the largest subset that can be fused without violating the FGW constraint. This is done via a binary search over the sorted list of children, requiring at most $\mathcal{O}(\log k)$ FGW evaluations. Each FGW computation is $\mathcal{O}(n^3)$, leading to a per-depth complexity of:

$$\mathcal{O}(\log k \cdot n^3)$$

**Edge Grafting with Dichotomic Search.** Similarly, for grafting, the algorithm considers up to $k^2$ grand-children (i.e., successors of successors), ranks them by similarity, and performs a binary search to connect the largest admissible subset without violating the FGW constraint. This again leads to at most $\mathcal{O}(\log k)$ FGW evaluations of cost $\mathcal{O}(n^3)$, for a total per-depth cost of:

$$\mathcal{O}(\log k \cdot n^3)$$

**Total Complexity.** Other operations, such as similarity computations, sorting, and updates of existence weights, are negligible compared to the FGW calls and bounded by $\mathcal{O}(k \cdot d)$ or $\mathcal{O}(k^2 \log k)$, which are typically much smaller than $\mathcal{O}(n^3)$.

The total complexity over $D$ recursive steps is therefore:

$$\boxed{\mathcal{O}(D \cdot \log k \cdot n^3)}$$

This complexity reflects the efficiency gained by using graph search over the space of fusion and grafting candidates rather than testing all combinations, while maintaining structural fidelity through FGW constraints. However, the constant $k$ is hard to estimate as it depends on the structure of the graph and thus the input document. However, we can assume that $k$ is upper bounded by a constant $k_{max}$, which is the

maximum number of children a node can have. Other models of complexity have been explored in the Appendix C. Those are based on some hypothesis on the structure of the document and lead to a closed-form expression of the constant $k$.

### 2.4 ENHANCED GRAPH AND QUESTION CLUSTERING WITH FGW

#### 2.4.1 OBJECTIVE

Outside of the context retrieval, The FGW distance can also be used as a way to cluster very specific questions, and more specifically what lies behind these questions thanks to the graph deformation. Indeed, one other method to weight the edges of the graph is to use the cosine similarity between the question and the content of the nodes, which can be computed as:

$$S_{qv} = \frac{\langle q, \chi_v \rangle}{\|q\|\|\chi_v\|}.$$

And then we can use the same transformation as before:

$$d_{qv}^p = \left( \frac{1}{\frac{\exp(S_{qv})}{\exp(1)}} \right)^p = \left( \frac{\exp(p)}{\exp(S_{qv} \cdot p)} \right).$$

Then we can use the FGW distance to compute the distance between the modified graphs obtained after the deformation involved by the different questions. The idea that lies behind this method is that if two questions deal with a close topic, even if the words used are different, the edge weights will be close enough to ensure a low FGW distance between them. This method can thus be used to cluster very specific questions and then the graphs obtained can be used to retrieve the relevant context for these questions.

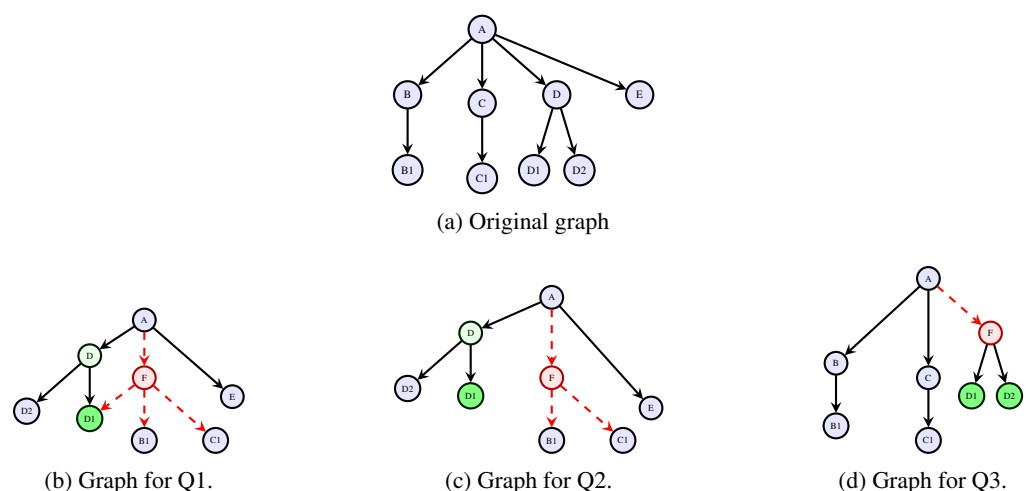

(a) Original graph

(b) Graph for Q1.    (c) Graph for Q2.    (d) Graph for Q3.

Figure 2: Top: original graph. Bottom: three variations. Q1 and Q2 fuse nodes B and C; Q3 fuses D and E.

In this example, there are three variations of the original graph. They are obtained from the original graph after asking three different questions, the Fused Gromov Wasserstein distance between the first two graphs will thus be low whereas the distance between the second and the third graph and the first and the third will be higher due to the different fusions of nodes and the different weights on the edges. We can use this metric

as an interesting way to cluster questions or texts based on their relation with the other parts of the texts. As explained, the FGW distance can be used to cluster questions related to a specific text based on the graph deformation that is induced by the questions.

### 2.4.2 CLUSTERING OF QUESTIONS AND AUTOMATED DOCUMENTATION GENERATION ON LARGE CORPUS

This perspective is interesting as it can allow us to discover the interesting clusters that are present in the text, and thus to discover the different topics that are present in the text. However, in order to do so, we would need to have a large number of questions related to the text. That is why we propose a two-way algorithm that first try to generate all the questions possible based on each branch of the graph, and then cluster these questions to discover the different relevant topics that are present in the text. This algorithm can be interesting to summarize the text and more specifically to create knowledge cards on the different topics presented in the text even if these topics are discussed in different parts of the text.

The clustering of questions can be used to generate automated documentation on large corpus, indeed it can help to identify the main topics of interest within the text. An algorithm that can generate such a documentation is presented in appendix D.

## 3 EXPERIMENTS

### 3.1 METHODOLOGY

We evaluated our approach on multiple datasets, such as HotPotQA (Yang et al., 2018), and LongBenchV2 (Bai et al., 2024). However, since our method relies on an evolving graph structure, we also conducted experiments to assess its adaptability to different types of questions and text variations on custom datasets. The interest of evaluating our method on these diverse datasets lies in its potential to generalize across various question types and text structures, while showing that it can still be used for classical retrieval and question answering tasks. We will mostly compare Kurisu-G² to the state-of-the-art methods such as RAG (Lewis et al., 2020) and other retrieval methods that works on Knowledge Graphs such as ToG-2.0 (Ma et al., 2024) or GraphRAG (Edge et al., 2024). HotpotQA is a multi-hop question answering benchmark in which each question requires combining information from multiple documents to produce the correct answer. It evaluates both the reasoning ability of the model and its robustness to noisy or partially relevant context. In contrast, LongBench-v2 focuses on evaluating models in long-context settings, with inputs reaching several tens of thousands of tokens. It includes diverse tasks such as multi-document understanding, summarization, and information retrieval. This benchmark is particularly relevant for assessing the effectiveness of Retrieval-Augmented Generation (RAG) methods in large-scale scenarios.

Model performance was evaluated using the Exact Match (EM) metric, defined as the percentage of predictions that exactly match the gold answer after normalization (removal of casing, whitespace, and punctuation). EM is a strict metric: any deviation, even minor, between the model output and the reference answer results in a score of zero for that example. This is why we used the F1 score as a secondary metric, which measures the overlap between the predicted and reference answers at the token level. However in order to compare to ToG-2.0, we used their script to compute the EM score since it is slightly more lenient than the classical EM score. For LongBenchV2, we used the multi-choice accuracy as the main metric, which is the percentage of questions for which the model selects the correct answer from a list of options.

### 3.2 BENCHMARK AND COMPARISON PROCESS WITH OTHER METHODS

We also evaluated our method on LongBenchV2, on the short (0-32k words) and medium tests (32k-128k words).We compared our method to the baseline RAG method.

Table 1: Performance comparison (Exact Match %) on HotpotQA

| Method | HotpotQA |
|---|---|
| Baseline RAG (Llama3-8B) | 24% |
| TOG-2.0 (Llama3-8B) | 34% |
| Kurisu-G2 (Llama3-8B) | **39%** |

Table 2: Performance comparison (Multi choices questions %) on LongBenchV2

| Method | Short | Medium |
|---|---|---|
| Baseline RAG (Llama3-8B) | 35% | 27.9% |
| Kurisu-G2 (Llama3-8B) | **38%** | **32%** |

However, the main benchmarks in the field of retrieval and question answering do not enable us to use the edge evolution mechanism well.

### 3.3 RUNTIME ANALYSIS

We conducted a runtime analysis to evaluate the efficiency of Kurisu-G² compared to other methods. Our experiments were performed on a machine with two Tesla T4 GPUs, and we measured the average time taken for each method to process a batch of 32 questions.

The results are summarized in Table 3. Kurisu-G² demonstrates competitive runtime performance, particularly in long-context scenarios where traditional methods struggle due to their reliance on LLMs to generate the Knowledge Graph. The implementation used for GraphRAG was adapted from the llama-index repository (Liu, 2022).

Table 3: Average runtime (in seconds) for a question using Llama3-8B.

| Method | Full Process | Inference |
|---|---|---|
| Kurisu-G2 | **70** | **25** |
| GraphRAG | 1200 | 175 |

## 4 DISCUSSION

Our experiments demonstrate that Kurisu-G² can outperform existing methods in retrieval while still being easy to implement and computationally efficient compared to other recent frameworks. However, there is still several areas for improvement and future work such as the creation of a specific benchmark that could measure how well the evolution of the graph performs.

For instance, there is room for improvement in the evolution phase of our algorithm, particularly in adapting to new information and user queries. Handling noisy data also remains an open question: while we avoided relying on large language models at the core of the algorithm, they could still help clean data or summarize nodes when documents become too large

### USAGE OF LARGE LANGUAGE MODELS

During this work, LLMs were used as a way to obtain ideas on how to polish some of the sentences in this article.

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

## A    ILLUSTRATION OF THE FUSED GROMOV-WASSERSTEIN DISTANCE

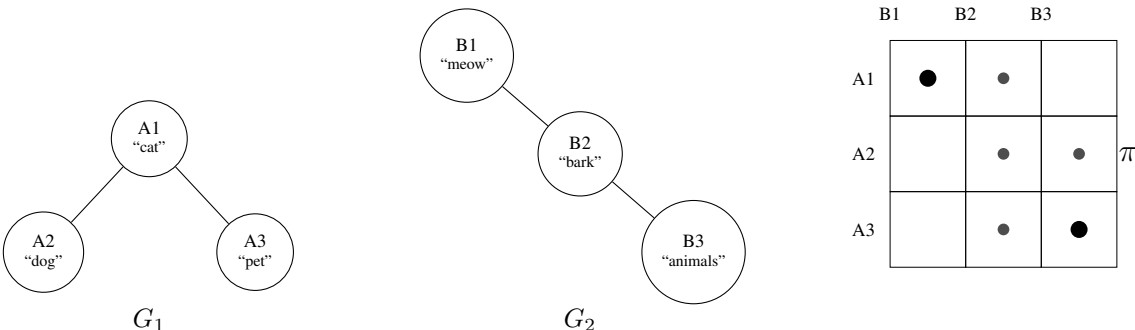

Figure 3: Example of a FGW computation between two graphs with different structures

## B    PSEUDO-CODE AND DISCUSSION ABOUT THE ALGORITHMIC DESIGN CHOICES

### B.1    PSEUDO-CODE OF THE ALGORITHM

---

**Algorithm 1** Recursive traversal with fusion and FGW constraints (Part 1)

---

1: **function** RECURSIVETRAVERSALFUSIONFGW($start\_node$, $question$, $G$, $embeddings$, $model$, $max\_depth$, $\tau$, $\varepsilon$, $\eta$)
2:     $q \leftarrow \text{Encode}(question)$
3:     $path \leftarrow [(start\_node, \text{None})]$
4:     $current \leftarrow start\_node$
5:     **for** $d = 0$ **to** $max\_depth$ **do**
6:         $C \leftarrow$ successors of $current$
7:         **if** $C = \emptyset$ **then**
8:             **break**
9:         **end if**
                                                        ▷ Compute similarities and filter relevant children
10:        Compute similarities $s_i = \langle q, v_{c_i} \rangle$
11:        $C_{\text{relevant}} \leftarrow \{c_i : s_i > \tau \cdot \max_j p_{current,c_j} \cdot s_j\}$
                                                        ▷ Greedy fusion phase
12:        $S \leftarrow \emptyset$
13:        **for** $c \in C_{\text{relevant}}$ **do**
14:            $S' \leftarrow S \cup \{c\}$
15:            $T' \leftarrow$ concatenation of the texts of the nodes in $S'$
16:            Create $G'$ with a fused node $\chi_d$ replacing $S'$
17:            **if** $\text{FGW}(G, G') \leq \varepsilon$ **then**
18:                $S \leftarrow S'$
19:            **else**
20:                **break**
21:            **end if**
22:        **end for**

---

---

**Algorithm 2** Recursive traversal with fusion and FGW constraints (Part 2)

---

23:        **if** $|S| > 1$ **then**
24:             Create $\chi_d$, replace $c \in S$, add $\chi_d$ to the graph
25:             Compute $p_{cf}$ as the mean of $p_{cu}$ for $u \in S$
26:             Assign $p_{cf}$ to the edge $(current, \chi_d)$
                                                                                  ▷ Edge grafting of additional edges
27:             $G_r \leftarrow$ grandchildren of $c \notin S$
28:             Sort $G_r$ by similarity with $q$
29:             **for** each $g \in G_r$ **do**                          ▷ In practice, this loop is dichotomic
30:                  Compute $\text{sim}(q, g)$
31:                  **if** $\text{FGW}(G, G + (\chi_d, g)) \leq \varepsilon$ **then**
32:                       Compute $p_{fg}^{(0)}$ (initial grafting weight)
33:                       Update $p_{fg}$:

$$p_{fg} \leftarrow \min(1, \ \max(0, \ p_{fg} + \eta \cdot (\text{sim}(\chi_d, g) - \tau)))$$

34:                  **end if**
35:             **end for**
36:             $current \leftarrow \chi_d$
37:             Append $(\chi_d, \text{score})$ to $path$
38:        **else**
39:             $current \leftarrow \arg\max s_i$
40:             Append $(current, s_{current})$ to $path$
41:        **end if**
42:   **end for**
43:   **return** $path$
44: **end function**

---

### B.1.1  PARAMETERS

- `start_node` : ID of the starting node;
- `question` : question used to measure similarity;
- `embeddings` : embedding vectors for each node;
- `G` : directed graph $G = (V, E)$;
- `model` : sentence encoder producing embeddings;
- `alpha` : weighting factor between content and structure in FGW;
- `max_depth` : maximum depth of exploration;
- `sim_threshold` : relative similarity threshold;
- `fgw_threshold` : maximum allowed FGW distance for a fusion.

### B.2  DISCUSSION ON ALGORITHMIC DESIGN CHOICES

A key design choice of the algorithm is to associate each edge with a tuple of labels: a structural dissimilarity and an existence weight. While the latter might seem optional, it is crucial for robustness. Without existence weights, a purely greedy strategy based only on structural dissimilarity risks discarding beneficial fusions, leading to overly rigid behavior. The existence weight acts as a soft balancing factor: it allows the algorithm to favor semantically relevant connections while still preserving the global structure.

For example, a node at the root of a large subgraph may have children highly relevant to the query, but strict FGW constraints could prevent their fusion due to structural deformation. Existence weights mitigate this by enabling the creation of new, semantically meaningful links without causing uncontrolled densification. In doing so, the algorithm avoids the main pitfall of purely fusion-based methods, where relevant knowledge remains disconnected. By dynamically adjusting these weights, it incrementally refines the graph's connectivity in a data-driven and query-sensitive way.

### B.3 EVOLUTION OF THE EXISTENCE WEIGHTS

**Reward distribution update.** Let $E$ be the set of candidate edges with similarities $s_e \in [0, 1]$. For a global reward budget $R > 0$, learning rate $\alpha \in (0, 1]$, and hyperparameters

$$\tau_+ > 0, \quad \tau_- > 0, \quad \eta \geq 0, \quad \delta > 0, \quad \gamma \in [0, 1), \quad \sigma_{\min} > 0,$$

we compute the following update.

**Centering and normalization.**

$$\mu = \frac{1}{|E|} \sum_{e \in E} s_e, \qquad \sigma = \sqrt{\frac{1}{|E|} \sum_{e \in E} (s_e - \mu)^2}, \qquad \tilde{\sigma} = \max(\sigma, \sigma_{\min}),$$

$$z_e = \frac{s_e - \mu}{\tilde{\sigma}}.$$

**Positive pool (boosts).** We shift $z_e$ by a margin $\eta$ to also include near-average edges:

$$p_e = \max\{0, \, z_e + \eta\}.$$

Softmax weights with temperature $\tau_+$ are:

$$w_e^+ = \frac{\exp(p_e/\tau_+)}{\sum_{j \in E} \exp(p_j/\tau_+)},$$

and the positive allocation is

$$a_e^+ = (1 - \gamma)R\, w_e^+, \qquad \sum_{e \in E} a_e^+ = (1 - \gamma)R.$$

**Negative pool (penalties).** For edges well below the mean:

$$M = \{e \in E \mid z_e < -\delta\},$$

we define

$$w_e^- = \begin{cases} \dfrac{\exp(|z_e|/\tau_-)}{\sum_{j \in M} \exp(|z_j|/\tau_-)} & e \in M, \\ 0 & e \notin M, \end{cases}$$

and set

$$a_e^- = -\gamma R\, w_e^-, \qquad \sum_{e \in E} a_e^- = -\gamma R.$$

**Final update rule.** Each edge evolves according to

$$p_e^{(t+1)} = \min\Big(1, \max\big(0,\ p_e^{(t)} + \alpha\,(a_e^+ + a_e^-)\big)\Big).$$

This update rule does not follow an existing formulation directly, but combines established principles: softmax-based allocation as in attention mechanisms (Vaswani et al., 2017), normalization via $z$-scores, reward shaping from reinforcement learning (Ng et al., 1999; Sutton & Barto, 2018), and projected updates under constraints

## C  DEEPER COMPLEXITY ANALYSIS

### C.1  AVERAGE-CASE COMPLEXITY WITH SIMILARITY-CONSTRAINED DECAYING CONNECTIVITY

We refine the average-case complexity by assuming that the average number of successors $k(d)$ at recursion depth $d$ decreases exponentially, with a decay rate $\beta$ that depends on the similarity threshold $\tau \in [0, 1]$. This threshold governs how selective the fusion and grafting operations are: higher values of $\tau$ restrict admissible connections and lead to sparser graphs.

**Connectivity Model.** We model the depth-dependent connectivity as:

$$k(d) = k_0 \cdot e^{-\beta(\tau)\cdot d}$$

with:

$$\beta(\tau) = \beta_0 + \gamma \cdot \tau \quad \text{where } \beta_0 \geq 0,\ \gamma > 0$$

This formulation reflects the intuition that:

- When $\tau$ is low (e.g., 0.2), many weakly similar units are connected, resulting in slower decay ($\beta(\tau) \approx \beta_0$).
- When $\tau$ is high (e.g., 0.8 or more), only highly similar nodes are merged or grafted, which leads to a sharp drop in connectivity with depth.

The parameter $k_0$ (or $k_{i,0}$ in the clustered model) represents the initial expected branching factor at the root level of the traversal. Its value is influenced by several factors, including the structural granularity of the input (e.g., paragraphs vs. sentences), the semantic density of the document or cluster, and the similarity threshold $\tau$ used to filter candidate successors. In general, higher content density or lower similarity thresholds tend to increase $k_0$, whereas stricter constraints or fragmented content reduce it.

**Complexity per Depth.** At each depth $d$, the number of FGW evaluations is $\log k(d)$, giving:

$$C(d) = \mathcal{O}(\log(k_0 e^{-\beta(\tau)d}) \cdot n^3) = \mathcal{O}((\log k_0 - \beta(\tau)d) \cdot n^3)$$

This is valid as long as $\log k_0 - \beta(\tau)d \geq 0$, i.e., $d < \frac{\log k_0}{\beta(\tau)}$. Define:

$$D' = \min\left(D, \left\lfloor \frac{\log k_0}{\beta(\tau)} \right\rfloor\right)$$

**Total Complexity.** The total cost becomes:

$$\sum_{d=0}^{D'} C(d) = \mathcal{O}\left(n^3 \cdot \left[(D'+1)\log k_0 - \frac{\beta(\tau)}{2}D'(D'+1)\right]\right)$$

$$\boxed{\mathcal{O}\left(n^3 \cdot \left[(D'+1)\log k_0 - \frac{(\beta_0 + \gamma\tau)}{2}D'(D'+1)\right]\right)}$$

## C.2 Alternative Complexity Model Based on Gaussian Clustering of Content

We propose an alternative probabilistic model of the algorithm's complexity, based on the empirical observation that documents often exhibit clustered structures, with semantically coherent sections containing varying amounts of relevant content. Rather than assuming uniform or exponentially decreasing connectivity, we model the distribution of relevant units as a mixture of Gaussian clusters.

**Document Structure as Thematic Clusters.** Let the input document be composed of $x$ semantic clusters (e.g., sections, argument blocks, or topics). Each cluster $i \in [1, x]$ is assumed to contain a random number $P_i$ of relevant subunits (e.g., paragraphs or phrases), drawn from a Gaussian distribution:

$$P_i \sim \mathcal{N}(\mu_i, \sigma_i^2), \quad P_i \geq 0$$

Here, $\mu_i$ denotes the expected number of relevant units (e.g., paragraphs or sentences) contained in cluster $i$, while $\sigma_i$ measures the variability of this number across different documents or instances of the same cluster type. These parameters describe the *size distribution* of the clusters. In contrast, $k_i(d)$ characterizes the *connectivity* within cluster $i$ at recursion depth $d$, i.e., the average branching factor of nodes once the cluster is instantiated. Thus, $\mu_i, \sigma_i$ control how many nodes are available in a cluster, whereas $k_i(d)$ determines how these nodes are connected and traversed during the algorithm.

We define the total number of relevant processing units as:

$$N = \sum_{i=1}^{x} P_i$$

Each cluster $i$ gives rise to a local subgraph where recursive traversal operates over the $P_i$ nodes. The average number of successors per node at recursion depth $d$ within cluster $i$ is modeled as:

$$k_i(d) = k_{i,0} \cdot e^{-\beta_i d} \quad \text{with } k_{i,0} \sim \mathcal{N}(\mu_i, \sigma_i^2)$$

This extends the exponential decay model by assigning cluster-specific initial connectivity and decay rates.

**Expected Complexity.** Assuming the dominant cost remains FGW computation $\mathcal{O}(n^3)$, the expected total cost over all clusters becomes:

$$\mathbb{E}[C_{\text{total}}] = \mathcal{O}\left(n^3 \cdot \sum_{i=1}^{x} \sum_{d=0}^{D_i} \mathbb{E}\left[\log\left(k_i(d)\right)\right]\right)$$

Using:

$$\log(k_i(d)) = \log(k_{i,0}) - \beta_i d \quad \Rightarrow \quad \mathbb{E}[\log(k_i(d))] = \mathbb{E}[\log(k_{i,0})] - \beta_i d$$

Hence:

$$\mathbb{E}[C_{\text{total}}] = \mathcal{O}\left(n^3 \cdot \sum_{i=1}^{x}\left[(D_i + 1) \cdot \mathbb{E}[\log(k_{i,0})] - \frac{\beta_i}{2}D_i(D_i + 1)\right]\right)$$

# D  AUTOMATIC DOCUMENTATION GENERATION ALGORITHM

Here is the pseudo-code of the algorithm that can generate automatic documentation on a large corpus based on the clustering of questions. It relies on a pre generations of basic questions about the document with an LLM, then a clustering of these questions in order to identify the main topics of interest, and finally a retrieval of relevant context with Kurisu-G2 and a generation of documentation with an LLM.

---

**Algorithm 3** GenerateAdequateDoc

---

**Require:** Large corpus $\mathcal{C}$ (documents $\rightarrow$ sections $\rightarrow$ paragraphs $\rightarrow$ sentences), LLM, clustering parameters `params`, Kurisu-G2 retrieval backend
**Ensure:** Structured documentation $\mathcal{D}$ guided by questions
 1: $G \leftarrow$ GENERATEDOCUMENTGRAPH($\mathcal{C}$)
 2: $\mathcal{Q} \leftarrow$ GENERATEQUESTIONSPERBRANCH($G$, LLM)
 3: $\mathcal{C}_q \leftarrow$ CLUSTERQUESTIONS($\mathcal{Q}$, `params`)
 4: $\mathcal{R} \leftarrow$ IDENTIFYCLUSTERSANDREPRESENTATIVEQUESTIONS($\mathcal{C}_q$, LLM)
 5: $\mathcal{D} \leftarrow \emptyset$
 6: **for all** $(i, q_i^\star) \in \mathcal{R}$ **do**
 7:     $\mathcal{X}_i \leftarrow$ KURISUG2RETRIEVECONTEXT($q_i^\star$)
 8:     $d_i \leftarrow$ LLMFORMATDOCUMENTATION($q_i^\star, \mathcal{X}_i$)
 9:     $\mathcal{D} \leftarrow \mathcal{D} \cup \{d_i\}$
10: **end for**
11: $\mathcal{D} \leftarrow$ MERGEDOCS($\mathcal{D}$)                    ▷ Deduplication, cross-links, table of contents
12: **return** $\mathcal{D}$

---

---

**Algorithm 4** GenerateDocumentGraph

---

 1: **function** GENERATEDOCUMENTGRAPH($\mathcal{C}$)
 2:     Initialize directed graph $G \leftarrow (V \leftarrow \emptyset, E \leftarrow \emptyset)$
 3:     **for all** document, section, paragraph, sentence in $\mathcal{C}$ **do**
 4:         create node $v$ with *id*, *text*, *embedding*
 5:         $V \leftarrow V \cup \{v\}$
 6:     **end for**
 7:     add hierarchical edges (doc→section, section→paragraph, paragraph→sentence)
 8:     add semantic edges between nodes with similarity > threshold
 9:     **return** $G$
10: **end function**

---

---

**Algorithm 5** GenerateQuestionsPerBranch

---

1: **function** GENERATEQUESTIONSPERBRANCH($G$, LLM)
2:     $\mathcal{B} \leftarrow$ TRAVERSEBRANCHES($G$)                       ▷ set of hierarchical paths
3:     $\mathcal{Q} \leftarrow \emptyset$
4:     **for all** branch $b \in \mathcal{B}$ **do**
5:         $q_b \leftarrow$ LLM.PROMPT("Generate $m$ relevant questions covering branch $b$")
6:         $\mathcal{Q} \leftarrow \mathcal{Q} \cup q_b$
7:     **end for**
8:     **return** $\mathcal{Q}$
9: **end function**

---

**Algorithm 6** ClusterQuestions

---

1: **function** CLUSTERQUESTIONS($\mathcal{Q}$, params)
2:     compute embeddings $E(q)$ for each $q \in \mathcal{Q}$
3:     Apply dimensionality reduction (e.g., PCA / t-SNE / MDS) to $E(q)$
4:     apply clustering algorithm (e.g., k-means) with `params`
5:     obtain clusters $\mathcal{C}_q = \{C_1, \ldots, C_K\}$
6:     **return** $\mathcal{C}_q$
7: **end function**

---

**Algorithm 7** IdentifyClustersAndRepresentativeQuestions

---

1: **function** IDENTIFYCLUSTERSANDREPRESENTATIVEQUESTIONS($\mathcal{C}_q$, LLM)
2:     $\mathcal{R} \leftarrow \emptyset$
3:     **for all** cluster $C_i \in \mathcal{C}_q$ **do**
4:         $q_i^\star \leftarrow$ LLM.PROMPT("Synthesize and rephrase a pivot question for this cluster:" $C_i$)
5:         $\mathcal{R} \leftarrow \mathcal{R} \cup \{(i, q_i^\star)\}$
6:     **end for**
7:     **return** $\mathcal{R}$
8: **end function**

---

**Algorithm 8** KurisuG2RetrieveContext

---

1: **function** KURISUG2RETRIEVECONTEXT($q^\star$)
2:     $\mathcal{X} \leftarrow$ KURISU-G2.RETRIEVE($q^\star$, top-$k$, filters, window)
3:     **return** $\mathcal{X}$
4: **end function**

---

**Algorithm 9** LLMFormatDocumentation

---

1: **function** LLMFORMATDOCUMENTATION($q^\star, \mathcal{X}$)
2:     **return**     LLM.PROMPT("Write a documentation section answering     $q^\star$     using only     $\mathcal{X}$ . Structure: title, summary, explanations, examples, cross-references.")
3: **end function**

---

**Algorithm 10** MergeDocs (optional post-processing)

```
1: function MERGEDOCS(D)
2:     deduplicate similar sections
3:     add cross-references and hyperlinks between related sections
4:     build a hierarchical table of contents
5:     return D
6: end function
```

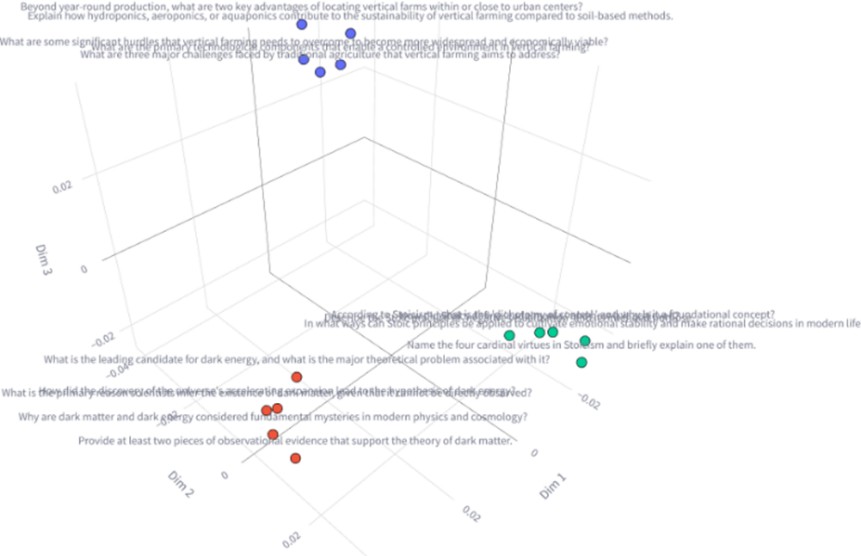

Figure 4: MDS clustering of questions based on Fused Gromov Wasserstein distance on graphs derivated from questions.

