# OpenReview forum: "Kurisu-G²: A Knowledge Retrieval and Generation Algorithm Based on Document Structure"
_ICLR.cc/2026/Conference — Submitted to ICLR 2026_

### Official Review · Reviewer_LqRF · 2025-10-15

**Soundness:** 2
**Presentation:** 2
**Contribution:** 3
**Rating:** 4
**Confidence:** 3

**Summary:**

The paper proposes KURISU-G², a RAG method that builds and traverses a Hierarchical Document Graph (HDG). Instead of LLM-constructed graphs (e.g., GraphRAG), it relies on a principled Fused Gromov–Wasserstein (FGW) distance to (i) greedily fuse relevant nodes and (ii) graft additional edges while enforcing a structural-consistency constraint. The goal is to form a coherent, query-conditioned subgraph that preserves both semantic and relational properties of the source document, thereby reducing redundant or incoherent context seen in vanilla chunk-ranking RAG and mitigating limitations of LLM-heavy graph construction.

**Strengths:**

S1. Clear HDG definition and FGW objective that jointly align content and structure during retrieval/fusion. This provides a theoretically grounded alternative to heuristic LLM-built graphs.

S2. Multiple perspectives (worst-case and average-case/clustered models) provide insight into scaling bottlenecks dominated by FGW computations.

S3. Higher EM on HotpotQA and multi-choice accuracy on LongBench-v2 vs baseline RAG, and much lower end-to-end runtime than GraphRAG in the authors’ setting.

**Weaknesses:**

W1. No ablations on key hyper-parameters, e.g., FGW weight α, similarity thresholds τ, grafting learning-rate/temperature parameters, and no study of how fusion set size affects quality/runtime.

W2. Only a few benchmarks and metrics. Lack of per-category analysis on HotpotQA.

w3. Section 2.2 includes too many formulas that are not essential to the core method, taking up an excessive amount of space.

**Questions:**

Firstly, please see my listed w1-w3 for details.

Further:

Q1. Since the paper emphasizes high-stakes domains, can you add experiments on domain-specific corpora (clinical guidelines, case law) and report robustness to noisy inputs?

q2. Could you compare to stronger structured-retrieval baselines that do not require LLM-constructed graphs (e.g., hyperlink/section-aware indexing + reranking), and to ToG-2.0 results on identical settings?

---

### Official Review · Reviewer_Jf14 · 2025-11-01

**Soundness:** 2
**Presentation:** 1
**Contribution:** 2
**Rating:** 2
**Confidence:** 4

**Summary:**

This paper presents KURISU-G2, a new graph-based Retrieval-Augmented Generation framework to tackle the issues of context fragmentation and incoherence. The crucial novelty is the use of the Fused Gromov–Wasserstein (FGW) distance for node fusion and graph grafting, independent of any LLM. The proposed model is benchmarked on HotpotQA and LongBenchV2, showing performance gains over the standard RAG baselines in addition to considerable computational efficiency improvements when compared to GraphRAG.

**Strengths:**

1. The main idea of using the FGW distance as a constraint for RAG context building is highly novel; it shifts the paradigm from heuristic, LLM-driven graph operations to a much more formal and mathematically solid method to preserve semantic and structural integrity.

2.  This is a well-written manuscript containing precise formal definitions of the core components. The proposed framework is well-articulated, and the analysis of its complexity is thorough.

3. This method provides consistent improvements on HotpotQA and LongBenchV2, reinforcing that structurally aware retrieval provides tangible benefits in downstream RAG performance.

**Weaknesses:**

1. The scalability analysis of FGW is missing. FGW is computationally expensive in principle; the paper does not yet convincingly demonstrate how the method scales to substantially larger graphs or document collections. A clearer description or empirical evidence of the approximations/optimizations used to achieve the reported runtimes is needed.

2. The writing needs to be improved. The manuscript states that “the main benchmarks... do not enable us to use the edge evolution mechanism well”. It is unclear whether Edge Evolution was used to produce the results in Tables 1 and 2. If it was not used, the mechanism should be decoupled from the main claim; if it was used, the paper must explain precisely how it was applied and evaluated.

3. Lack of hyperparameter sensitivity analysis. The FGW threshold (ε) appears to be a critical parameter, but the paper lacks a sensitivity analysis or ablation showing how EM/F1 and fusion behavior vary with ε. Without that analysis, it is hard to assess robustness—too large a threshold enables spurious fusions; too small prevents beneficial ones.

4. The experiment design is too simple. The authors only compare to GraphRAG and ToG-2.0. A lot of the latest baseline methods are missing, such as:

- [1] Reasoning on Graphs: Faithful and Interpretable Large Language Model Reasoning.
- [2] GNN-RAG: Graph Neural Retrieval for Large Language Model Reasoning.
- [3] ArchRAG: Attributed Community-based Hierarchical Retrieval-Augmented Generation.
- [4] PathRAG: Pruning Graph-based Retrieval Augmented Generation with Relational Paths.
- [5] NodeRAG: Structuring Graph-based RAG with Heterogeneous Nodes.

5. While runtime comparisons to GraphRAG are provided, the paper omits direct performance comparisons with GraphRAG and other GraphRAG-based methods. Moreover, it lacks evaluations against alternative similarity or fusion algorithms and different backbone models.

6. Although the authors provided the code in the supplementary materials, it is not complete, and the key scripts and implementation details are missing.

**Questions:**

NA

---

### Official Review · Reviewer_usJS · 2025-11-01

**Soundness:** 1
**Presentation:** 1
**Contribution:** 2
**Rating:** 2
**Confidence:** 4

**Summary:**

The paper addresses limitations in current Retrieval-Augmented Generation (RAG) methods, which tend to treat retrieved knowledge chunks independently, neglecting their semantic and logical dependencies and leading to incoherent or incomplete outputs. The recent GraphRAG approach introduces graph structures to encode chunk relationships, but has drawbacks: LLM-dependent graph construction, high computational cost, static graphs, and limited logical coherence. The authors propose Kurisu-G², a novel retrieval and generation algorithm that leverages the Gromov–Wasserstein (FGW) distance to guide context retrieval on a document-derived graph. This principled, mathematically-grounded measure preserves both structural relations and semantic similarity when merging/fusing knowledge units. The retrieval process recursively traverses and fuses nodes in the document graph, uses constrained graph modifications, and dynamically evolves edge weights.

**Strengths:**

The paper provides extensive theoretical descriptions and clear visualizations.

**Weaknesses:**

- The writing and formatting of this paper do not meet ICLR standards.
- The description of the proposed method is hard to understand and does not convincingly show that Kurisu-G² can improve the performance over GraphRAG. Moreover, document-derived graphs have already been explored in many baselines, such as RAPTOR.
- The experimental results are insufficient. There are only a few datasets, and baselines are evaluated, which is not enough to demonstrate the effectiveness of the proposed approach.

**Questions:**

As outlined under Weaknesses.

---

### Official Review · Reviewer_GfzZ · 2025-11-02

**Soundness:** 1
**Presentation:** 3
**Contribution:** 3
**Rating:** 2
**Confidence:** 5

**Summary:**

This paper proposes a new direction that avoids using LLMs or completely relying on LLMs for a high-quality graph in such a concentrated way. The applied FGW is clearly defined. However, the experiments of this paper are far from convincing to address the concerns. It still lasks pretty much work to prove the feasibility.

**Strengths:**

- The use of the FGW distance provides a mathematically sound framework to balance semantic relevance with structural consistency during retrieval.
- The observations suggest a wider applicability of the framework. But to me, this is more suitable for memory establishment, rather than the document generation example in the paper.

**Weaknesses:**

There are no sufficient experiments to make this paper technically sound, where a lot of effort should be put to address my concerns.
- There are no ablation studies to tell the importance of FGW constraint, fusion, grafting, and weight evolution.
- Scalability issue. may be the method could perform well on a smaller corpus, but it will face a severe efficiency problem on larger ones.
- Efficiency problem. Similar to the former one, 70s per question is definitely not acceptable for either a researcher or an industrial implementation.
- Unfair comparison. The runtime comparison between GraphRAG and this paper is calculated including the KG construction. Though existing methods rely on a pre-constructed graph, they are static and can be readily used for all the questions. However, this paper will construct a graph for each question each time, which makes this comparison unfair and unnecessary.
- Lack of baselines. The performance is indeed bad since the compared baselines are already reported to be the n-to-last ones. HippoRAG 1&2 and RAPTOR should be included.
- The framework relies on many hyperparameters, this make the paper less robust.

**Questions:**

- What is the typical value of n during the FGW computation? This could affect the scalability of the framework.
- How could the hyperparameters be tuned across benchmarks?

---

### Meta-Review · Area_Chair_v4aZ · 2026-01-06

**Summary:**

We have four reviewers who have carefully checked this paper and they all lean towards rejection. The main concern is the insufficient experiments. Also, the writing needs to be improved for clarity. No rebuttal was provided. Thus, I recommend that the paper should be revised and resubmitted with more comprehensive revision.

**Reviewer Concerns:**

No rebuttal was provided.

**Reviewer Scores:**

No rebuttal was provided.

---

### Decision · Program_Chairs · 2026-01-26

Reject